# Microwave-Assisted Metal-Organic Frameworks Derived Synthesis of Zn_2_GeO_4_ Nanowire Bundles for Lithium-Ion Batteries

**DOI:** 10.3390/nano13081432

**Published:** 2023-04-21

**Authors:** Chaofei Guo, Shuangqiang Chen, Junaid Aslam, Jiayi Li, Li-Ping Lv, Weiwei Sun, Weimin Cao, Yong Wang

**Affiliations:** 1Department of Chemical Engineering, School of Environmental and Chemical Engineering, Shanghai University, 99 Shangda Road, Shanghai 200444, China; chaofeiguo@shu.edu.cn (C.G.); chensq@shu.edu.cn (S.C.); junaidaslam1993@shu.edu.cn (J.A.); liping_lv@shu.edu.cn (L.-P.L.); vivisun@shu.edu.cn (W.S.); 2Department of Chemistry, College of Sciences, Shanghai University, 99 Shangda Road, Shanghai 200444, China; 1541435235@shu.edu.cn

**Keywords:** lithium-ion batteries, metal-organic frameworks, nanowire bundles, Zn_2_GeO_4_

## Abstract

Germanium-based multi-metallic-oxide materials have advantages of low activation energy, tunable output voltage, and high theoretical capacity. However, they also exhibit unsatisfactory electronic conductivity, sluggish cation kinetics, and severe volume change, resulting in inferior long-cycle stability and rate performance in lithium-ion batteries (LIBs). To solve these problems, we synthesize metal-organic frameworks derived from rice-like Zn_2_GeO_4_ nanowire bundles as the anode of LIBs via a microwave-assisted hydrothermal method, minimizing the particle size and enlarging the cation’s transmission channels, as well as, enhancing the electronic conductivity of the materials. The obtained Zn_2_GeO_4_ anode exhibits superior electrochemical performance. A high initial charge capacity of 730 mAhg^−1^ is obtained and maintained at 661 mAhg^−1^ after 500 cycles at 100 mA g^−1^ with a small capacity degradation ratio of ~0.02% for each cycle. Moreover, Zn_2_GeO_4_ exhibits a good rate performance, delivering a high capacity of 503 mA h g^−1^ at 5000 mA g^−1^. The good electrochemical performance of the rice-like Zn_2_GeO_4_ electrode can be attributed to its unique wire-bundle structure, the buffering effect of the bimetallic reaction at different potentials, good electrical conductivity, and fast kinetic rate.

## 1. Introduction

Rechargeable lithium-ion batteries (LIBs) have been widely used in fields such as electronics, power grids, and electric vehicles, due to their excellent performance. However, the low capacity (~372 mAh g^−1^) of the graphite anode material of LIBs cannot meet the current market demand for high-energy storage [1,2,3,4]. Therefore, there is an urgent need to develop high-energy-density electrode materials to meet the needs of social development [5]. Anode materials such as Si, Ge, Sn, Zn, Ni and their oxides have high theoretical capacities. As a result, they have been intensively studied by many researchers [6,7,8,9,10]. In the above electrode materials, Ge-based electrode materials exhibit a high theoretical capacity of ~1600 mAh g^−1^ due to the reversible reaction of each Ge atom with 4.4 lithium atoms. Moreover, Ge also possesses fascinating features, such as high conductivity and fast Li^+^ diffusivity, making it a promising anode candidate [11,12]. However, the volume of the germanium-based anode changes drastically during the frequent embedding and de-embedding of lithium ions, which seriously affects its structural stability [13]. Decreasing the particle size or dimensions to the nanoscale (such as nanoparticles, nanorods, nanofibers, nanowires, etc. [14,15]) can shorten the lithium ions’ diffusion pathway and, to some degree, mitigate physical strains, thereby lowering the capacity fading. Moreover, the reserve of Ge on earth is not abundant, which makes it far from a practical application because of its cost and elemental abundance [16]. The design of bimetal oxides is an excellent option for decreasing material costs and maintaining the low discharging platform. The design of the optimized morphology and other elements such as Zn and O to form Ge-based bimetal oxides have been proven effective for improving the materials’ electrochemical performances [17].

Despite great efforts, the practical application of this Zn_2_GeO_4_-based anode is still plagued by its rapid capacity decay and poor rate performance. Combining Zn_2_GeO_4_ with carbon materials or other materials such as graphene, carbon nanowires, or carbon nanotubes is an effective way to solve the problem. Zhang et al. successfully synthesized Zn_2_GeO_4_-C-Ag by a simple impregnation/redox method. The electrode exhibited better electrochemical performance, and the specific capacity could be stabilized at 561 mAh g^−1^ after 50 cycles. In addition, the nanoscale Zn_2_GeO_4_ nanorods were prepared through a hydrothermal method by Feng et al. This electrode material exhibits good electrochemical performance with an initial capacity of up to 995 mAh g^−1^ at 400 mA g^−1^ [18,19].

Lately, metal-organic framework (MOF) derived materials have gradually developed into promising anode materials for LIBs [20,21,22]. The broad investigation and utilization of MOFs are attributed to the highly-tunable and skeletal structures, and high surface-to-volume ratios, which contribute to the structural stability and electrical conductivity [23,24,25]. Therefore, MOFs with 3D pore architecture are rapidly developed and widely applied in energy storage, gas absorption, electrochemical, etc. [26,27,28]. Inspired by the merits arising from MOFs and Ge-based bimetal compounds, we propose the microwave-assisted method synthesis of Zn_2_GeO_4_ anodes for LIBs using MOFs as both structural and compositional precursors. The fabricated Zn_2_GeO_4_ could consequently inherit its MOF precursor’s morphology and porous structure, which is supposed to facilitate lithium diffusion and improve the structural stability. In addition, the Zn_2_GeO_4_ prepared by a microwave hydrothermal method has high phase purity and is suitable for the preparation of small-sized Zn_2_GeO_4_ crystals, showing great advantages over the hydrothermal/solvent thermal method. The microwave heating method involves the interaction between the electromagnetic radiation and the molecular dipole moment, and this method allows the temperature of the whole reaction system to be uniform and greatly improves the reaction rate for the preparation of Zn_2_GeO_4_ with uniform shapes compared with the conventional hydrothermal/solvent heat method.

In this work, to improve the electron transferability of germanium-based polymetallic oxide materials, accelerate the transport rate of lithium ions in the electrode, and improve the lithium storage performance, we synthesized rice-like Zn_2_GeO_4_ nanowire bundles as the anode for LIBs by the microwave-assisted hydrothermal method. The obtained Zn_2_GeO_4_ possesses minimized particle sizes, expanded cation transport channels, improved material electronic conductivity, and excellent electrochemical properties. The rice-like Zn_2_GeO_4_-150 nanowire bundles delivered a high reversible capacity of 730 mAh g^−1^ and maintained it at 661 mAh g^−1^ after 500 cycles at a current density of 100 mA g^−1^. The superior electrochemical performances of Zn_2_GeO_4_-150 nanowire bundles are likely ascribed to the large specific surface area, robust structure, and the buffering effect of bi-metal reactions at different potentials. Since the Zn_2_GeO_4_ prepared in this paper has good long-cycle stability and a fast kinetic rate, it is expected to be a lithium-ion battery anode with high-energy-density and long cycle life.

## 2. Materials and Methods

### 2.1. Chemicals

Zn(NO_3_)_2_⋅6H_2_O (99.7%, Shanghai, China), methanol (99.7%, Shanghai, China), 2-methyl imidazole (MIz, 98%, Shanghai, China), 1,2-diaminocyclohexane (DACH, 98%, Shanghai, China), GeO_2_ (99.99%, Shanghai, China) were bought from Sinopharm Chemical.

### 2.2. Synthesis of Zn-MOF (Zeolitic Imidazolate Framework-8, ZIF-8)

Under stirring conditions, 1.1 g of Zn(NO_3_)_2_ 6H_2_O was dissolved in 40 mL of methanol, labeled as solution A. Under stirring conditions, 2.4 g of MIz was dissolved in 40 mL of methanol and labeled as solution B. Subsequently, solution B was added to solution A and continued stirring for 2–3 h. Then, the product was collected by centrifugal washing with methanol several times and dried under vacuum at 60 °C.

### 2.3. Synthesis of Ge-MOF

Under stirring conditions, 0.2 g of GeO_2_ was dispersed in 10 mL of deionized water, and then 7.5 g of DACH was dropped into the GeO_2_ suspension. Then, the above solution was transferred to a 50 mL Teflon-lined stainless steel autoclave and heated at 180 °C for 5 days. After cooling to room temperature, the solution was rinsed with acetone and deionized water, and dried under vacuum at 60 °C for 12 h to obtain a white precipitate of Ge-MOF.

### 2.4. Synthesis of Zn_2_GeO_4_

An amount of 0.16 g of NaOH was dissolved into 30 mL of deionized water under stirring conditions. Then 0.16 g of ZIF-8 and 0.16 g of Ge-MOF were added into the above solution with stirring for 30 min. Finally, the mixed solution was transferred to a glass reaction vessel and allowed to react for 10 min under continuous magnetic stirring at 150 °C under microwave irradiation. The white precipitate was collected and centrifuged with deionized water several times and dried under a vacuum at 60 °C overnight. Then, the precursor was treated in a tube furnace at 700 °C for 5 h protected by N_2_ to form a final product, which was named Zn_2_GeO_4_-150. For comparison, Zn_2_GeO_4_-130 and Zn_2_GeO_4_-180 were also prepared at 130 °C and 180 °C for 10 min under microwave irradiation and treated at 700 °C for 5 h.

### 2.5. Material Characterization

The scanning electron microscope (SEM, JSM-6700F, Tokyo, Japan) with energy dispersive spectroscopy (EDS) was used to characterize the morphology and microstructure of the products with a microscope acceleration voltage of 30 KV. Transmission electron microscopy (TEM, JEOL JEM-200CX, Tokyo, Japan) was used to characterize the internal structure of the products with a microscope acceleration voltage of 200 KV. X-ray diffraction (XRD, Rigaku D/max-2550 V, Cu Ka radiation, Tokyo, Japan) was used to determine the crystals of the product with a test voltage and current of 40 KV/30 MA, a test range of 5–85°, and a scan speed of 2°/min. X-ray photoelectron spectroscopy (XPS, PHI-5700, Tokyo, Japan) was used. The equipment of micromeritics instrument Corp, ASAP 2020 was used to measure the Brunauer-Emmett-Teller (BET) surface area of samples at 77 K.

### 2.6. Electrochemical Evaluation

The material in the working electrode consists of the active material, acetylene black, and poly(vinylidene fluoride) (PVDF), with a mass ratio of 8:1:1. The mass load of electrode material on copper foil is maintained at around 2 mg cm^−2^, and maintained a thickness of several micrometers. Lithium foil was used as the anode for assembling coin batteries; LiPF_6_ is a solvent for using propyl carbonate/ethyl carbonate (PC/EC) as an electrolyte. Constant current charging and discharging are measured at room temperature between 5 mV and 3.0 V (relative to Li^+^/Li) at different current densities (0.1–5 C) on a LAND-CT 2001, 1 C = 1000 mA g^−1^. The voltage ampere (CV) circuit was measured at a scanning rate of 0.1 mV S^−1^ on a CHI660D. Electrochemical resistance testing (EIS) was performed on CHI660D at a frequency range in 0.01–1 × 10^6^ Hz.

## 3. Results

### 3.1. Characterization of Anode Materials

The synthesis process of the MOF derived rice-like Zn_2_GeO_4_ nanowire bundles is schematically illustrated in Figure 1. XRD was performed on the Zn_2_GeO_4_ products, Ge-MOF, and Zn-MOF. The typical XRD pattern of Ge-MOF is shown in Figure 2a, where all peaks are well-matched with the previously reported sample of C_6_H_10_(NH_2_)_2_Ge_3_O_6_ [29,30]. Furthermore, all the prominent peak positions of ZIF-8 are in good agreement with the previous reports [31,32]. Figure 2b shows that the prominent peaks of Zn_2_GeO_4_, including (110), (300), (220), (113), (410), and (223), are in good agreement with the rhombohedral Zn_2_GeO_4_ (JCPDS No. 11-0687), indicating a high purity of Zn_2_GeO_4_ via the microwave-induced method [33]. All the Zn_2_GeO_4_-130, -150, and -180 samples exhibited type IV nitrogen adsorption/desorption isotherms with typical H1 hysteresis loops and their specific surface areas were approximately ~31, ~43 and ~26 m^2^ g^−1^, respectively (Figure 2c,d), and the major pore sizes of all three Zn_2_GeO_4_ samples ranged from ~3 to 13 nm. The large difference in the specific surface area of Zn_2_GeO_4_ is mainly due to the difference in sample morphology and crystallinity caused by the synthesis temperatures of various samples.

Because of its highest surface area, the Zn_2_GeO_4_-150 sample was chosen for further investigations of the chemical composition and chemical bonding state by XPS analysis (Figure 3a–d). Figure 3a shows the overall XPS spectrum of the Zn_2_GeO_4_-150. The high-resolution XPS spectra of the binding energies at 1045 and 1023 eV in Figure 3b correspond to the Zn 2p_3/2_ and Zn 2p_1/2_ peaks, showing the oxidation state of Zn^2+^ [34,35]. Figure 3c,d reveal the binding energies of Ge 3d and O 1s peaks, centered at 32 and 531.2 eV, respectively, which prove the existence of Ge-O bonding in the form of tetravalent Ge^4+^ [36]. In addition, the O 1s split peaks appear at 530.3 eV, 531.0 eV and 532.0 eV, corresponding to lattice oxygen, oxygen of the hydroxide ion and C-O or O-C=O bonds, respectively. All the binding energy results are consistent with the XRD results of Zn_2_GeO_4_ [37].

The surface morphology and structure are shown in Figure 4a–h by SEM and TEM images. The Ge-MOF precursor with a plate-like morphology is shown in Figure 4a. A rhombic dodecahedral morphology of the Zn-MOF sample is shown in Figure 4b, with an average size of ~50 nm. As seen in Figure 4c, Zn_2_GeO_4_-150 particles are uniformly distributed with a rough surface, uniform size, and a rice grain shape with a diameter of ~90 nm and a length of ~250 nm. A closer observation is presented in Figure 4d–f, rice-like Zn_2_GeO_4_ nanowire bundles were accumulated by several nanowire bundles, which were further confirmed by the TEM image (Figure 4g,h). Figure 4g,h presents a mass of small nanowires with an average width of ~20 nm and a length of ~80 nm, mounted on the surface of rice-like Zn_2_GeO_4_-150 particles. The rice-like Zn_2_GeO_4_ nanowire bundles are accumulated by several nanowire bundles which may mitigate the volume change during cycling and promote a long-cycle stability of the electrode. In addition, the small size of the Zn_2_GeO_4_ nanowire bundle particles can shorten the ion and electron transport paths and promote fast reaction kinetics in the electrode. The EDS spectrum shown in Figure 4i demonstrates the presence of Zn, Ge, and O elements. In addition, the C element is also observed, which is derived from the carbon-based adhesive tape substrate, used to support and disperse the SEM samples. The SEM image reveals the structure of Zn_2_GeO_4_ under the different microwave irradiation temperatures of 130 and 180 °C in Figure 4j,k. It is confirmed that the single crystallite of Zn_2_GeO_4_-130 (Figure 4j) is not yet formed at low temperatures. As the microwave irradiation temperature increases up to 180 °C, the rice-like structure of Zn_2_GeO_4_ breaks down and is transformed into a random rectangular structure (Figure 4k). Figure 4l, l-a–l-c shows multiple element mapping images that confirm the uniform distribution of Ge, Zn, and O elements.

### 3.2. Electrochemical Performances as Anode Material for LIBs

Rice-like Zn_2_GeO_4_-150 and other Zn_2_GeO_4_ samples (Zn_2_GeO_4_-130 and Zn_2_GeO_4_-180) were all investigated as anode materials for LIBs at room temperature. The electrochemical properties are shown in Figure 5a–c. The electrochemical reaction mechanism of Zn_2_GeO_4_-150 was performed by investigating its initial three cyclic voltammograms (CV). During the first cathodic scan, a sharp peak was detected at around 0.6 V, which can be attributed to the formation of a solid electrolyte interface (SEI) film, and the Zn_2_GeO_4_ decomposed into Zn, Ge, and Li_2_O. Another peak ranges from 0.3 V to 0.005 V, which corresponds to the formation of Li_x_Zn and Li_y_Ge alloys [38,39,40]. The Li_2_O matrix, formed during the discharging process, can be used as a cushioning pad to alleviate the volume changes of Ge and Zn during the Li_x_Zn and Li_y_Ge. alloying/de-alloying processes [41,42]. In the first anodic scan, two remarkable broad peaks were observed at ~0.45 and 1.2 V, ascribed to the delithiation of Li_x_Zn and Li_y_Ge alloys at 0.45 V, and then further partially oxidized to the ZnO and GeO_2_ at 1.2 V [43]. For the subsequent anode scan, two major reduction peaks were observed in the range of 0.6 to 1.2 V and 0.005 to 0.6 V. The major 0.75 V peak in the second cycle is shifted to a higher voltage compared to the peak in the first cycle and then returns to a lower voltage in the third cycle. This phenomenon suggests that the subsequent redox reactions occur mainly in the Li_2_O matrix with Zn-ZnO and Ge-GeO_2_, while the reactions between 0.6 and 0 V are due to the alloying of Li with Zn and Ge. In the subsequent cycles, the observed CV peaks almost overlap, indicating the excellent reversible electrochemical behavior of Zn_2_GeO_4_ [44]. The initial capacity loss may come from the formation of an SEI layer and the multi-step reactions of Zn_2_GeO_4_ [45,46]. Finally, during the discharging and charging processes, all electrochemical reactions can be described by the following Equations (1)–(5) [39,40,41,42,43,44,45,46]:Zn_2_GeO_4_ + 8Li^+^ + 8e^−^ → 2Zn + Ge + 4Li_2_O(1)
Zn + xLi^+^ + xe^−^ ↔ Li_x_Zn (0 ≤ x ≤ 1)(2)
Ge + yLi^+^ + ye^−^ ↔ Li_y_Ge (0 ≤ y ≤ 4.4)(3)
ZnO + 2Li^+^ + 2e^−^ ↔ Zn + Li_2_O(4)
GeO_2_ + 4Li^+^ + 4e^−^ ↔ Ge + 2Li_2_O(5)

The cycle performances of Zn_2_GeO_4_-130, Zn_2_GeO_4_-150, and Zn_2_GeO_4_-180 electrodes are shown in Figure 5c. Among them, Zn_2_GeO_4_-150 exhibited the best electrochemical performance. A high initial reversible capacity of 730 mA h g^−1^ was obtained at a current density of 100 mA g^−1^ and remained at 661 mAh g^−1^ after 500 cycles, showing a high-capacity retention ratio of 90.5% and a low capacity degradation of about 0.02% for each cycle. In the initial cycle, there is an irreversible capacity loss, which is mainly due to the decomposition of the electrolyte and the formation of a solid electrolyte interface (SEI) layer. Moreover, a stable Coulombic efficiency (close to 100%) was obtained after the second cycle and remained stable in subsequent cycles. The excellent cyclic performance of the Zn_2_GeO_4_-150 electrode is probably ascribed to the porous structure, the formation of Li_x_Zn and Li_y_Ge alloys with rich redox reactions and favorable buffering effects and enhanced electrical conductivity [47,48]. The two samples of Zn_2_GeO_4_-130 and Zn_2_GeO_4_-180 delivered higher reversible capacities of 700 mAh g^−1^ and 690 mAh g^−1^, respectively, close to the initial capacity of Zn_2_GeO_4_-150. However, capacities of Zn_2_GeO_4_-130 and Zn_2_GeO_4_-180 faded rapidly, with the retained capacities of ~410 mA h g^−1^ and 390 mA h g^−1^ respectively after 500 cycles, followed by the low-capacity retention ratios.

The rate capabilities of Zn_2_GeO_4_-130, Zn_2_GeO_4_-150, and Zn_2_GeO_4_-180 are shown in Figure 6a. Even though the current densities were changed rapidly, the Zn_2_GeO_4_-150 sample still exhibited the most stable cycling performance and the highest capacity. As shown in Figure 6a, the average specific capacities of the rice-like Zn_2_GeO_4_-150 were 730, 698, 627, 587, 544, and 503 mAh g^−1^ at step current densities of 0.1 to 5 C. When the current density returned to 0.1 C, the capacity of Zn_2_GeO_4_ almost recovered (up to 709 mAh g^−1^), demonstrating the reversible insertion-extraction of Li cations and the stable rice-like structure [49]. In contrast, Zn_2_GeO_4_-130 and Zn_2_GeO_4_-180 showed unfavorable rate capacities with a low and rapid decrease in capacities under the same testing conditions, revealing the significant impacts of synthesis temperatures on the electrochemical performance of samples. The large difference in rate capacity between Zn_2_GeO_4_-130 and Zn_2_GeO_4_-180 is mainly due to the effect of the microwave synthesis temperature on the crystalline structure of the Zn_2_GeO_4_ material, which in consequence affects the electrical conductivity, ion diffusion rate, and structural stability of the material.

The obtained Nyquist plots of the AC impedance of the Zn_2_GeO_4_ products are shown in Figure 6b. The Nyquist plots of Zn_2_GeO_4_-150 have a similar semi-circle, along with the Zn_2_GeO_4_-130 and Zn_2_GeO_4_-180 samples. However, the charge transfer resistance (R_ct_) of Zn_2_GeO_4_-150 is ~75 Ω. This is much smaller than Zn_2_GeO_4_-130 (~150 Ω) and Zn_2_GeO_4_-180 (~100 Ω), demonstrating that the rice-like Zn_2_GeO_4_-150 has better electronic conductivity and lower inner electrochemical resistance. In particular, compared to previous Ge-based anodes (Table 1) with all kinds of morphologies, such as nanoflower, nanorod, nanowire, or nanofiber, the rice-like Zn_2_GeO_4_-150 nanowire bundle electrode stands out by its remarkable electrochemical performance, which is ascribed to the robust structure and proper porosity derived from MOF precursors.

As the excellent cyclic performance and rate performance of Zn_2_GeO_4_-150 draw our attention, the relative lithium kinetics in Zn_2_GeO_4_-150 should also be measured to explain the possible reason. Figure 7a shows the CV of the Zn_2_GeO_4_-150 electrode at different scan rates, from 0.2 to 1.0 mV s^−1^. Peak 1 corresponds to the redox reactions occurring mainly in the Li_2_O matrix with Zn-ZnO and Ge-GeO_2_ and peak 2 corresponds to the reaction of the Li_x_Zn and Li_y_Ge alloys oxidized to the ZnO and GeO_2_ at 1.2 V. The lithium storage mechanism could be revealed by the relationship between the measured current (*i*) and the scan rate (*v*) from the CV curves, using the equations as follows: $i=a\times v^{b}$, $\log\left(i\right)=b\times \log\left(v\right)+log(a)$, where (*v*) represents the scan rate, (*i*) represents the current; both a and b (ranges from 0.5 to 1) are constants [51]. Figure 7b,c reveal the relationship between $\log\left(i\right)$ and $\log\left(v\right)$ of the Zn_2_GeO_4_-150. It showed that the values of b are 0.71 and 0.57 for oxidation along with a reduction state, respectively, suggesting both conversion reaction and capacitive behavior occur during the charge-discharge processes. Furthermore, the capacitive capacity was calculated using the equation: $i_{\left(v\right)}=k_{1}\times v+k_{2}\times v^{1/2}$, where *i*_(*v*)_ represents the measured current at a fixed potential (*v*), $k_{1}$ and $k_{2}$ are the adjustable constant parameters, and v represents the scan rate [50]. The capacitive capacity contribution is about 57.9% for the Zn_2_GeO_4_-150 electrode at 0.2 mV s^−1^. Particularly, when the value of *v* was increased up to 1.0 mV s^−1^, the capacitive contribution enhanced up to 76.6% (Figure 7d). The capacitive contribution of the Zn_2_GeO_4_-150 electrode increases with the increase in the scan rate, which indicates the large existence of surface absorption mechanism and copious amounts of active spots of the rice-like Zn_2_GeO_4_ nanowire bundles.

## 4. Conclusions

In summary, this work synthesized the rice-like Zn_2_GeO_4_ nanowire bundles using a microwave-induced hydrothermal method with subsequent pyrolysis of the MOF precursor. The Zn_2_GeO_4_-150 nanowire bundle electrode delivers a high initial reversible capacity of 730 mAh g^−1^ at a current density of 100 mA g^−1^ and retains a high reversible capacity of 661 mA h g^−1^ after 500 cycles. Moreover, good high-rate capabilities (1685, 698, 627, 577, 544, 503 mAh g^−1^ at step-up current densities of 0.1 C to 5 C) were also achieved because of the tiny particle size and improved conductivity as well as the robust structural stability of the Zn_2_GeO_4_-150 anode. The Zn_2_GeO_4_-150 nanowire bundles have also shown superior cycling performance with a high capacity retention ratio of 90.5% (500 cycles), which is attributed to the composition and structure associated merits such as the large specific surface area, robust carbon structure, buffering effect of the bi-metal reactions at different potentials, and excellent electronic conductivity, as well as the fast kinetics.

## Figures and Tables

**Figure 1 nanomaterials-13-01432-f001:**
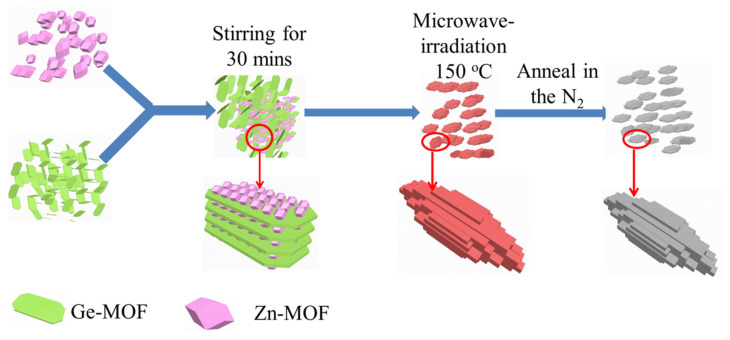
Schematic diagram of the microwave-assisted synthesis of metal-organic frameworks derived rice-like Zn_2_GeO_4_ nanowire bundles.

**Figure 2 nanomaterials-13-01432-f002:**
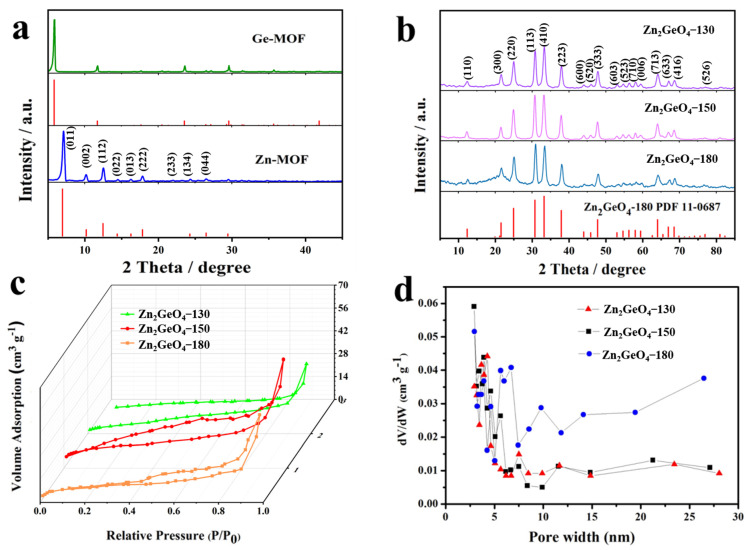
(**a**) XRD curves of Ge-MOF and Zn-MOF, (**b**) XRD curves of Zn_2_GeO_4_-130 and Zn_2_GeO_4_-150 and Zn_2_GeO_4_-180. (**c**) Nitrogen adsorption/desorption isotherm of the Zn_2_GeO_4_-130 and Zn_2_GeO_4_-150 and Zn_2_GeO_4_-180, (**d**) the corresponding pore size distribution curves.

**Figure 3 nanomaterials-13-01432-f003:**
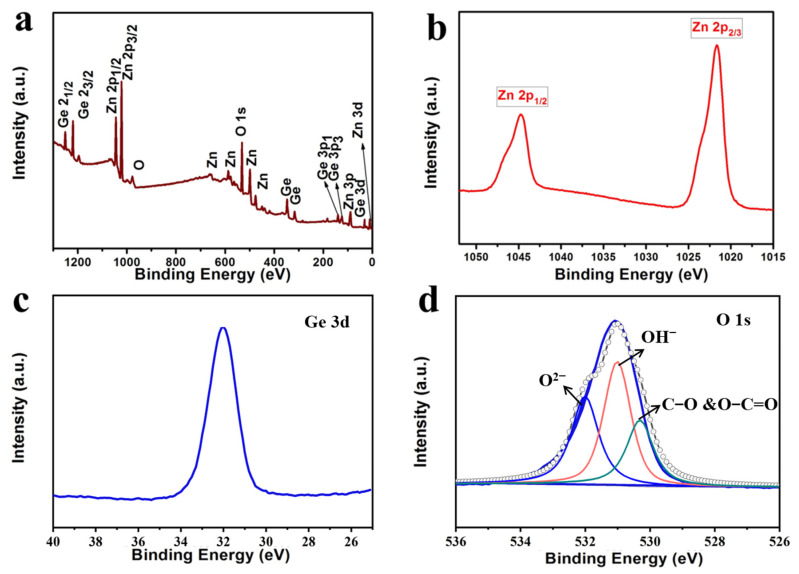
XPS spectra of Zn_2_GeO_4_-150: (**a**) survey spectrum, (**b**) Zn 2p peak, (**c**) Ge 3d peak, (**d**) O 1s peak.

**Figure 4 nanomaterials-13-01432-f004:**
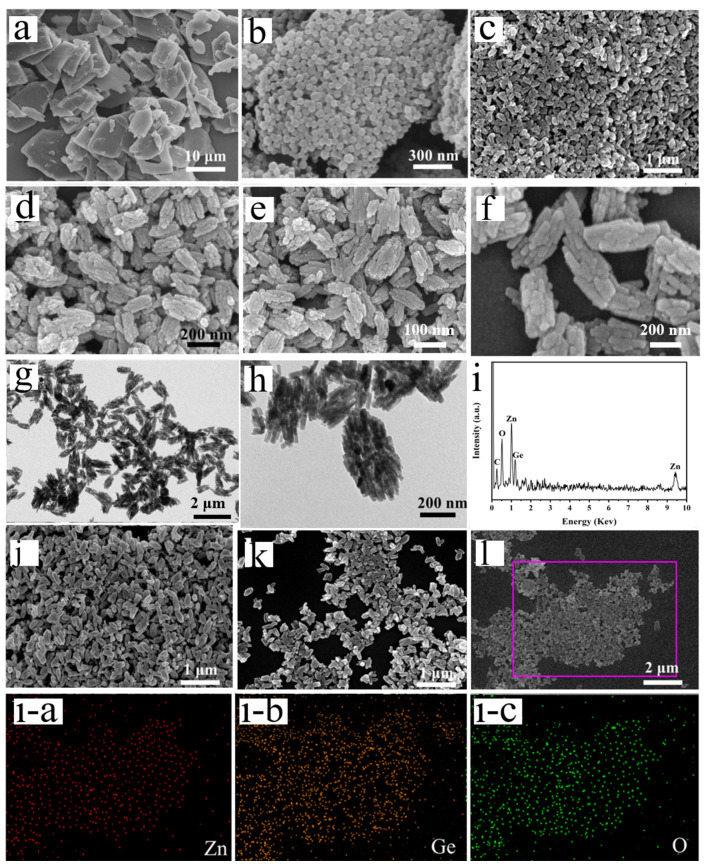
SEM images of (**a**) Ge-MOF, (**b**) Zn-MOF and (**c**–**f**) Zn_2_GeO_4_-150. (**g**,**h**) TEM image of Zn_2_GeO_4_-150 sample. (**i**) EDS profile of Zn_2_GeO_4_-150. (**j**) SEM image of Zn_2_GeO_4_-130, (**k**) SEM image of Zn_2_GeO_4_-180, and (**l**) SEM image of Zn_2_GeO_4_-150 and the corresponding elemental mapping images of (**l-a**) Zn, (**l-b**) Ge and (**l-c**) O.

**Figure 5 nanomaterials-13-01432-f005:**
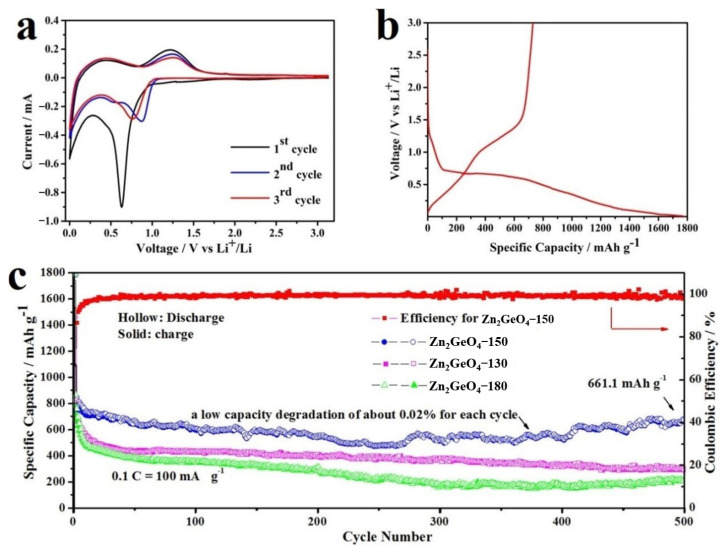
(**a**) CV curves of Zn_2_GeO_4_-150 electrode at 0.1 mV s^−1^. (**b**) The discharge/charging curves of Zn_2_GeO_4_-150 electrode at 100 mA g^−1^. (**c**) Cycling performances of Zn_2_GeO_4_-130, Zn_2_GeO_4_-150 and Zn_2_GeO_4_-180 at 100 mA g^−1^, and Coulombic efficiencies of the main product of Zn_2_GeO_4_-150.

**Figure 6 nanomaterials-13-01432-f006:**
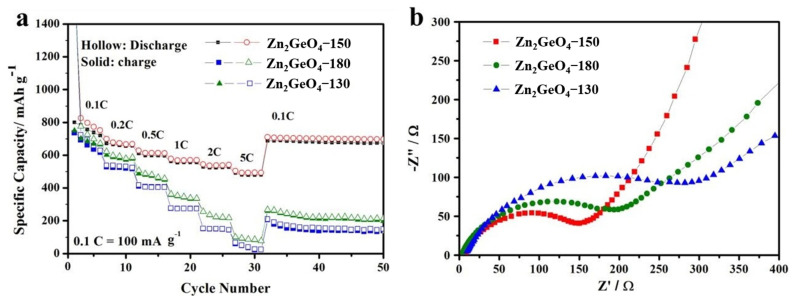
(**a**) Rate performances of Zn_2_GeO_4_-130, Zn_2_GeO_4_-150 and Zn_2_GeO_4_-180 at different current rates. (**b**) EIS spectra of three electrodes.

**Figure 7 nanomaterials-13-01432-f007:**
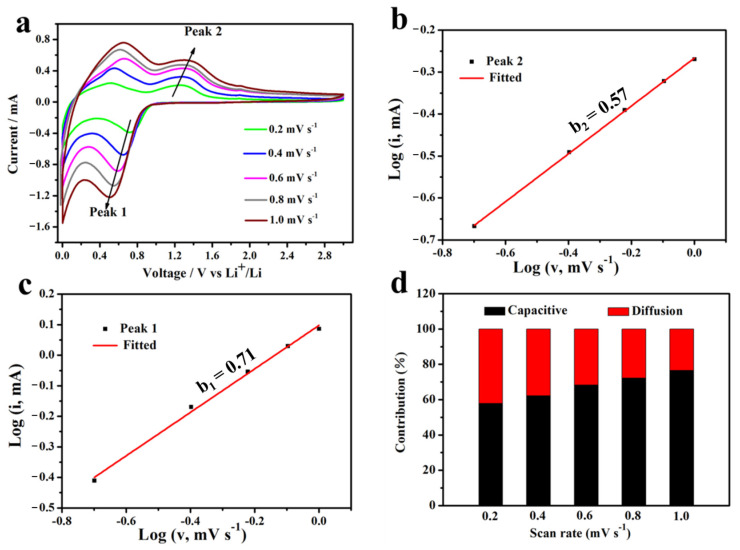
(**a**) CV curves of Zn_2_GeO_4_-150 electrode at different scan rates ranging from 0.2 to 1 mV s^−1^, (**b**,**c**) the corresponding plots of $\log\left(i\right)$ versus $\log\left(v\right)$, (**d**) contribution ratio of capacitive and diffusion-controlled behaviors at various scan rates.

**Table 1 nanomaterials-13-01432-t001:** Comparison of electrochemical properties of Zn_2_GeO_4_ and relative metal oxide composites. (IRC: initial reversible capacity, mAh g^−1^; RRC: retained reversible capacity, mAh g^−1^; CN: cycle number; CD: current density, mA g^−1^; V: voltage, V).

Composite	Morphology	IRC	RRC/CN	CD	V	References
Zn_2_GeO_4_ Ge/Zn_2_GeO_4_NFs	rice flower	730 1621	661/500 816/200	100 200	0.005–3 0.01–3	This work [5]
Zn_2_GeO_4_	nanoflower	1143	1034/160	500	0.01–2.8	[12]
Zn_2_GeO_4_@MWCNTs	nanorods@voids of MWCNTs	1209	1397/300	200	0.01–3	[13]
Zn_2_GeO_4_	nanoparticle	~1130	1175/60	200	0.01–3	[17]
Zn_2_GeO_4_	nanowire	600	485/900	600	0–3	[29]
Zn_2_GeO_4_	nanowires	2200	1200/150	100	0.01–3	[33]
Zn_2_GeO_4_/GO	nanorod@sheets	594	1150/100	200	0.001–3	[36]
Zn_2_GeO_4_	nanospheres	1520	488/100	200	0.01–3	[37]
Zn_2_GeO_4_	nanofiber	1405	1084/50	200	0.01–3	[38]
Zn_2_GeO_4_@C/Cu	ZGO@C nanowires	1162	~790/100	200	0.01–3	[39]
Zn_2_GeO_4_/TiO_2_	rod-like microstructure	346	330/150	200	0.01–3	[41]
Zn_2_GeO_4_/RGO	hollow rods@sheets	1736	1005/100	500	0.01–3	[45]
CNT-Zn_2_GeO_4_	3D CNT@microspheres	736	762/300	150	0.001–3	[48]
Zn_2_GeO_4_	nanowire	1135	1220/100	100	0.01–3	[50]

## Data Availability

The data presented in this study is available on request from the corresponding authors.

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
