# Peer review of "Microwave-Assisted Metal-Organic Frameworks Derived Synthesis of Zn2GeO4 Nanowire Bundles for Lithium-Ion Batteries"

_nanomaterials, 2023, doi:10.3390/nano13081432_

Round 1
Reviewer 1 Report
Germanium-based multi-metal oxide materials have advantages for low activation energy, tunable output voltage and high theoretical capacity. Many researchers are improving cyclic stability and rate performance by overcoming electronic conductivity, sluggish cation kinetics and serious volume change. In this paper, Zn2GeO4 nanowire bundles were synthesized as an anode of a lithium-ion battery through a microwave hydrothermal method to expand the transmission channels of cations and improve their electronic conductivity.
The authors noted that among the composites of Zn2GeO4-130, Zn2GeO4-150 and Zn2GeO4-180, the Zn2GeO4-150 sample gave the best performance. It is necessary to clearly present on which criteria Zn2GeO4-150 has excellent performance. Also, in Table 1, the Zn2GeO4-150 sample was compared with other references, and the performance difference in terms of morphology was described. Why do you think there is a difference in performance depending on the morphology?
Minor
1. Complete the text with sentences in the 2.1 chemical section.
2. Use the correct mathematical operators.
3. Check the units and subscripts.
4. In Figure 4 (I) create a sub-list in the mapping image to the SEM image. (e.g. I-a, I-b, I-c)
Pay attention to SI units and subscripts, and write sentences clearly. Readers will find it a little more readable.
Author Response
Thanks for your valuable comments. Please, see the attachment.

Reviewer 2 Report
The present work describes development of a LIB anode material based on zink germanate. The introduction is clear and comprehensive, all the relevant references are provided. The methods are mostly well described and the results are quite interesting. In my opinion, the work is fit for publication after some most minor changes listed below:
1) Please check the manuscrpt for typos and minor mistakes. For example, in line 25 and 27 the degree symbol is missing, and in some other places the font is mixed up.
2) The last paragraph of introduction (lines 60-70) is more suitable for the conclusion, as it summarizes the results of the paper. I would suggest to rewrite it and state the goal of the work and the means to achieve it in this section instead.
3) In line 87 the abbreviation ZIF-8 is used for the first time and its meaning must be explained.
4) In section 2.5 please provide more data, such as the accelerating voltage for the microscopy, step aquisition time and step size for XRD etc. In line 98 it is likely that the device for BET surface measurements is not named correctly.
5) BET surface area measurements have typical error of about 20%. Please provide numbers with appropriate accurace in line 123 and further in the text.
6) Please make the pictures in Fig.2 bigger. Provide hkl indexes on the Fig. 2a, 2b. If possible, remake the curves in Fig. 2c and 2d to be offset, otherwise they overlap too much.
7) Please remove grey edges from all the figures.
8) Please mention the phase composition of the product after the MW treatment.
9) Please describe Fig.4f some more. Some of the peaks are not attributed.
10) In line 162 and 190 there is obviosly a typo in the sample names.
I would advise to go through the work once more and fix some of the phrasing (e.g. in line 257 and other places). The flaws are minor and the overall English is nice and comprehensive.
Author Response

(The authors gave the same response as above.)

Reviewer 3 Report
The reported microwave synthesis is a relatively new chemical method to facilitate reactions and could be another avenue for green synthesis of nanomaterials. Several attributes of microwave heating contribute to greener syntheses, including shorter reaction time, lower energy consumption, and higher product yield. Therefore, this technique qualifies the requirement of being a fast synthetic method to produce radically advanced electrode materials for energy storage applications. Many synthetic approaches have been reported for the preparation of well-defined Zn2GeO4-based architectures. However, in the presented work, MOF derived with improved energy density and power capability of Zn2GeO4-based nanocomposites with unique functions is desirable. The paper itself is written well and seems worthwhile but the main concern of this paper and the areas that require clarification are given below.
· In the introduction, the authors should briefly explain the conventional heating process for synthesizing nanomaterials and how it differs from microwave dielectric heating.
· Does the MOF materials ZIF-8 used to coat the surface of the Zn2GeO4 electrode? Is this like a core/shell structure?
· Binary metal oxides exhibit specific functions and have been used as anode electrodes for LIBs, however, in the introduction (lines 40 – 41) other anodes such as Zn and Ni (reported in the domain doi.org/10.1021/acsami.0c13755) must also be included.
· The rationale for using DACH in section 2.3 should be detailed.
· The clarity of Figure 2a must be improved. All XRD peaks must be labeled.
· What is the overall conclusion from the XPS results?
· The rice to rectangular shape is hard to view through SEM images.
· In equations (1) – (5); why is there no reaction for forming LixGe?
· A large irreversible capacity between the discharge and charge in the initial cycle must be explained. Was their formation of a solid electrolyte interface (SEI) layer, similar to that in previously reported metal-oxide anodes that were based on conversion- and alloy-type cycling mechanisms, please address this.
· The role of ZIF-8 on the surface of ZGO nanoparticles should be justified.
· The contribution of capacitance at various scan rates can be correlated to the mechanism reported in the literature (doi.org/10.1016/j.ceramint.2022.03.266).
· Page 8, Lines 209 – Line 211: Is the observed difference due to the synthesized material's conductivity?
· In Figure 7a, what do peaks 1 and 2 correspond to?
Some minor edits will do.
Author Response

(The authors gave the same response as above.)

Reviewer 4 Report
The authors have successfully prepared Zn2GeO4 nanowire bundles microwave technique for Li batteries applications. The idea and the quality of analysis are excellent. Also, the authors wrote and organized the manuscript thoroughly. In this regard, I have only a few comments to improve the quality of the article for the reader and to meet the nanomaterials journal standard.
1. In BET results please explain why different growth temperature showed different specific surface area.
2. In XPS result, the O1s and Ge3d peaks deconvoluting into several peaks at different binding energies for different bonding.. It is important from authors to fit O1s and Ge3d peaks.
3. Please summary in table the composition of the Zn2GeO4 obtained from SEM.
4. Table 1. Please compare your performance with more recent references. Currently, the used references is old.
English language fine. No issues detected
Author Response

(The authors gave the same response as above.)

Round 2
Reviewer 3 Report
The revised version is suitable for publication.
The language is fairly OK
Reviewer 4 Report
The authors improved the manuscript remarkably and addressed all my comments. I can recommend this manuscript for the publication in nanomaterials journal.
Moderate editing of English language